# Your best day: An interactive app to translate how time reallocations within a 24-hour day are associated with health measures

Dorothea Dumuid[1,2]*, Timothy Olds[1,2], Melissa Wake[2,3,4], Charlotte Lund Rasmussen[5,6], Željko Pedišić[7], Jim H. Hughes[8,9], David JR. Foster[10], Rosemary Walmsley[11,12], Andrew J. Atkin[13], Leon Straker[14], Francois Fraysse[1], Ross T. Smith[15], Frank Neumann[16], Ron S. Kenett[17,18], Paul Jarle Mork[5], Derrick Bennett[12,19,20], Aiden Doherty[11,12,19], Ty Stanford[1]

1 Alliance for Research in Exercise, Nutrition and Activity, Allied Health & Human Performance, University of South Australia, Adelaide, South Australia, Australia, 2 Murdoch Children's Research Institute, Parkville, Victoria, Australia, 3 Department of Paediatrics, The University of Melbourne, Parkville, Victoria, Australia, 4 Liggins Institute, The University of Auckland, Grafton, New Zealand, 5 Department of Public Health and Nursing, Norwegian University of Science and Technology, Trondheim, Norway, 6 Faculty of Physical Culture, Palacký University, Olomouc, Czech Republic, 7 Institute for Health and Sport, Victoria University, Melbourne, Victoria, Australia, 8 School of Pharmacy and Medical Sciences, University of South Australia, Adelaide, South Australia, Australia, 9 Pfizer Inc., Groton, CT, United States of America, 10 Clinical and Health Sciences, Australian Centre for Precision Health, University of South Australia, Adelaide, South Australia, Australia, 11 Big Data Institute, Li Ka Shing Centre for Health Information and Discovery, University of Oxford, Oxford, United Kingdom, 12 Nuffield Department of Population Health, University of Oxford, Oxford, United Kingdom, 13 School of Health Sciences, University of East Anglia, Norwich, United Kingdom, 14 School of Allied Health, Faculty of Health Science, Curtin University, Perth, Australia, 15 Australian Centre for Interactive and Virtual Environments, Wearable Computer Lab, University of South Australia, Adelaide, South Australia, Australia, 16 Optimisation and Logistics, School of Computer Science, The University of Adelaide, Adelaide, South Australia, Australia, 17 The KPA Group, The Samuel Neaman Institute, Technion, Haifa, Israel, 18 University of Turin, Turin, Italy, 19 National Institute of Health Research Oxford Biomedical Research Centre, Oxford University Hospitals NHS Foundation Trust, John Radcliffe Hospital, Oxford, United Kingdom, 20 Medical Research Council Population Health Research Unit, University of Oxford, Oxford, United Kingdom

* dot.dumuid@unisa.edu.au

**Data Availability Statement:** Data cannot be shared publicly because restrictions apply to access the data. The data that support the findings

## Abstract

Reallocations of time between daily activities such as sleep, sedentary behavior and physical activity are differentially associated with markers of physical, mental and social health. An individual's most desirable allocation of time may differ depending on which outcomes they value most, with these outcomes potentially competing with each other for reallocations. We aimed to develop an interactive app that translates how self-selected time reallocations are associated with multiple health measures. We used data from the Australian Child Health CheckPoint study ($n$ = 1685, 48% female, 11–12 y), with time spent in daily activities derived from a validated 24-h recall instrument, %body fat from bioelectric impedance, psychosocial health from the Pediatric Quality of Life Inventory and academic performance (writing) from national standardized tests. We created a user-interface to the compositional isotemporal substitution model with interactive sliders that can be manipulated to self-select time reallocations between activities. The time-use composition was significantly associated with body fat percentage (F = 2.66, $P$ < .001), psychosocial health (F = 4.02, $P$ < .001), and academic performance (F = 2.76, $P$ < .001). Dragging the sliders on the

of this study are available from the National Centre for Longitudinal Data (NCLD) but restrictions apply to the availability of these data, which were used under license for the current study, and so are not publicly available. To become a licensed data user for the CheckPoint and LSAC studies, researchers are required to sign a confidentiality deed, found here: https://ada.edu.au/wp-content/uploads/2019/11/Confidentiality-Deed-Poll-November-2019.pdf. The confidentiality deed imposes a legal obligation to not share any unit-level data with anyone unless they also are registered data users. Researchers can apply to become registered data users by following the link provided below. https://growingupinaustralia.gov.au/data-and-documentation/accessing-lsac-data. Researchers are able to access these data in the same manner as the authors. The authors did not have any special access privileges.

**Funding:** DD is supported by the Australian National Health and Medical Research Council (NHMRC) Early Career Fellowship APP1162166 and by the Centre of Research Excellence in Driving Global Investment in Adolescent Health funded by NHMRC APP1171981. MW is supported by NHMRC Principal Research Fellowship APP1160906. AD is supported by the Wellcome Trust [223100/Z/21/Z] and the National Institute for Health Research (NIHR) Oxford Biomedical Research Centre (BRC). RW is supported by a Medical Research Council Industrial trategy Studentship (grant number MR/S502509/1). This study was supported by NHMRC Ideas APP1186123. The CheckPoint study was supported by the NHMRC [APP1041352; APP1109355]; the National Heart Foundation of Australia [100660; The Royal Children's Hospital Foundation [2014-241]; the Murdoch Children's Research Institute (MCRI) [No award number available]; The University of Melbourne [No award number available]; the Financial Markets Foundation for Children [2014-055, 2016-310]; and the Australian Department of Social Services (DSS) [No award number available]. Research at the MCRI is supported by the Victorian Government's Operational Infrastructure Support Program [No award number available]. The funders played no role in the study design, data collection and analysis, decision to publish, or preparation of the manuscript.

**Competing interests:** The authors have declared that no competing interests exist.

app shows how self-selected time reallocations are associated with the health measures. For example, reallocating 60 minutes from screen time to physical activity was associated with -0.8 [95% CI -1.0 to -0.5] %body fat, +1.9 [1.4 to 2.5] psychosocial score and +4.5 [1.8 to 7.2] academic performance. Our app allows the health associations of time reallocations to be compared against each other. Interactive interfaces provide flexibility in selecting which time reallocations to investigate, and may transform how research findings are disseminated.

## Introduction

How we use our time may affect our health, wellbeing and productivity [1–6]. Daily time allocations to sleep, sedentary behavior and physical activity are associated with adiposity, cardio-metabolic biomarkers, mental health and cognition/academic performance in adults [1–3] and children [4–6]. When we change the time spent in an activity, this must always be accompanied by a net equal and opposite compensatory change across the remaining activities to maintain the daily 24-hour total [7]. The characteristic interdependency between individual time-use components poses both analytical and interpretational challenges (e.g. is a change in outcome driven by more time sleeping or less time studying to compensate for that extra sleep?). Novel application of statistical methods such as compositional data analysis overcome these challenges [8, 9].

Compositional data analysis calculates isometric log-ratio coordinates (S1 File) of time-use data. These log-ratios are then used as independent variables in regression models for health outcomes [10]. This avoids perfect multicollinearity between the parts of a full time-use composition, while also respecting the inherent relative nature of the data [11]. However, the regression coefficients from compositional regression models, representing the associations with log-ratio coordinates, can be difficult to interpret in practical terms. Log-ratio coordinates are not expressed in min/day (or other such time unit); instead their regression coefficients represent the estimated difference in the outcome variable when the log-ratio coordinate increases by one. This means relationships between time use and outcomes can be challenging to interpret for health professionals and the general population. Consequently, people may be precluded from choosing or promoting data-driven choices regarding the use of time. Thus, accessible tools are needed to enable simple interpretations of how time-use reallocations relate to health.

Compositional isotemporal substitution analysis overcomes the interpretability issue and provides practically meaningful and straightforward interpretation of the regression coefficients for log-ratio coordinates, by quantifying the health associations of reallocating absolute durations (e.g. 30 minutes) of time between daily activities [12]. However, there are many different possible reallocations that could be explored and reported (e.g., different durations, between any number of different activities). Two types of reallocation are commonly studied: *one-for-one* (reallocation of time from one activity to one other activity) and *one-for-remaining* (reallocation of time from one activity to all the remaining activities *pro rata*) [11].

However, these types of reallocations may not always be relevant to real life situations. From the limited evidence available, it appears that people do not directly swap one activity for another (one-for-one), nor do they compensate for increasing one activity by changing all remaining activities (one-for-remaining) *pro rata* [13, 14]. For example, when adults retire, they predominantly distribute the time previously spent at work to sleep, screen time and

chores and not to moderate-vigorous physical activity only [15]. For greater relevance to intervention design and real life, more flexibility is required when defining time reallocations for isotemporal substitution models.

Estimates from compositional isotemporal substitution models are calculated with respect to representative "starting" time-use compositions [12]. To date, most studies have used the average composition (i.e., compositional centre, see S1 File) observed in the sample as the starting composition from which time between activities is reallocated [16]. However, we have previously demonstrated the results differ when an alternative starting composition is used [17], and the benefits of reallocation may be greatest for those furthest from the average (e.g. the most sedentary individuals). Allowing time reallocations to be modelled from different starting compositions may extend the usefulness and relevance of the findings to specific population groups or individuals.

Accurate time-use data from very large population-based cohorts and biobanks linked with a vast array of clinical and biological data are becoming increasingly available and allow more precise predictive modelling. Drawing on these data resources and the advanced statistical methods outlined above, user-friendly interactive tools are needed to understand and utilize the richness of the data.

The aim of this study was, therefore, to develop an interactive web application that could be used to communicate the potential impact of user-defined time-use reallocations on a selection of health measures. In this study, three child health measures ─ body fat percentage, psychosocial health, and academic performance ─ were selected, but the methodology presented herein is applicable to other age groups and health measures/outcomes.

## Materials and methods

### Study design and participants

The data used to underpin the app are from the Child Health CheckPoint [18], a cross-sectional study conducted within the Longitudinal Study of Australian Children (LSAC) [19]. In 2003 LSAC recruited 5107 infants, to create a nationally representative sample from all Australian states and territories, and followed them up in biennial waves. In 2015, between waves 6 and 7, children were invited to participate in an additional biophysical assessment module. Of those children still enrolled in LSAC at wave 6 ($n$ = 3764, 74% retention), 1874 (42%, i.e. 37% of the original LSAC sample) consented to participate in the Child Health CheckPoint study. At age 11–12 years, children attended a single centre-based or home-based assessment. The data collection took place between February 2015 and March 2016, with additional data collected by phone in the following weeks.

The research was carried out according to The Code of Ethics of the World Medical Association (Declaration of Helsinki). The Child Health CheckPoint study was approved by the Royal Children's Hospital Melbourne Human Research Ethics Committee (ref: 33225D) and the Australian Institute of Family Studies Ethics Committee (ref: 14–26). A parent or guardian provided written informed consent for their child's participation in the study. Data for this study were drawn from the CheckPoint dataset released to all research end-users via the ADA Dataverse [20].

### Measures

Daily activity composition was derived from a computerized 24-h recall called the Multimedia Activity Recall for Children and Adolescents (MARCA) [21]. Children recalled the activities (e.g. sleeping, cycling, watching TV) they did on the previous day at a granularity of 5 minutes. The MARCA instrument maps up to 500 activities into the following superdomains: sleep,

screen time, physical activity, quiet time, passive transport, school-related activities, and domestic/self-care activities. Three days, including at least one school and one non-school day, were recalled, the first during a face-to-face interview at the initial assessment visit, and the other two via a telephone interview during the subsequent week. Time use across the three days was weighted for weekdays:weekend days at 5:2.

For the purposes of this paper, the illustrative health measures were body fat percentage, psychosocial health, and academic performance in a writing test. Body fat percentage was estimated using bioelectric impedance, measured in bare feet and light clothing to the nearest 0.1% with a 4-limb InBody230 (Biospace, Seoul, South Korea) at centre visits, or a 2-limb Tanita BC-351 (Kewdale, Australia) at home visits [22]. The Psychosocial Health Summary Score from the 8- to 12-year-old self-report Pediatric Quality of Life Inventory (PedsQL), version 4.0 [23] was calculated from the emotional, social, and school functioning domains. Higher scores indicated better psychosocial health. Academic performance was obtained from standardized Year 7 National Assessment Program for Numeracy and Literacy (NAPLAN) tests [24], carried out when the children were approximately 12–13 years old. We used the writing component scores, trimmed at +/- 4 standard deviations (SD) to remove outliers and normalize the distribution. In the writing tests, students were given a topic and asked to write a response in either narrative writing or persuasive writing. Tests were marked centrally against the Australian Curriculum core capabilities, assessing criteria such as the capacity to engage the audience, formulation of ideas, vocabulary, cohesion, punctuation, spelling and text structure.

Covariates were selected due to their known associations with children's time-use behaviours and health outcomes. They included child's sex, age, pubertal status, and family-level socioeconomic position. Sex and age were obtained from parent report. Pubertal status (pre-pubertal, early pubertal, mid-pubertal, late pubertal or post-pubertal) was derived from children's self-reported pubertal signs, using the Pubertal Development Scale [25] during the CheckPoint study. A composite *z*-score from LSAC Wave 6 incorporating parent-reported occupation, household income, and highest parental education level was used as a measure of family-level socioeconomic position [26].

## Statistical analysis and development of the shiny app

The compositional isotemporal substitution analyses were conducted in R version 4.1.0 [27] using the "compositions" [28] and "zCompositions" [29] packages. Following published procedures, the seven-part 24-hour activity composition (sleep, screen time, physical activity, quiet time, passive transport, school-related activities, and domestic/self-care activities) was expressed as a set of six isometric log ratios, after all zeros in the raw time-use data were replaced by non-zero values using a log-ratio expectation-maximization algorithm [30]. Three multiple linear regression models were fit, one for each of the health measures (i.e. body fat percentage, psychosocial health and academic performance). Participants with missing data were excluded from the analytical samples. The body fat percentage outcome variable was log (ln)-transformed to normalize the right-skewed distribution of residuals. The time-use composition isometric log ratios were used as the predictor variables along with the covariates (sex, age, pubertal status, and family-level socioeconomic position). In an effort to improve the model fits and account for non-linear relationships between time-use components and the outcomes, second-order polynomial (interactions and squared) isometric log-ratio terms were included [31]. The Akaike information criterion (AIC) was reduced for all models when the polynomial predictor variables were included [32].

The regression parameters from the compositional models were extracted to be used in the app. The open-source R statistical software [27] Shiny package [33] enables the programming

of an interactive interface that can be freely accessed via a web-link, without the need to install R or any other software. The Shiny app developed in the current study has three components (i.e. R scripts) that communicate with each other: (1) the user interface (ui.R), which defines the app's appearance and how the user enters information; (2) the server (server.R), which monitors and makes computations on input provided by the user interface to then in turn provide results back to the user interface; and (3) the global script (global.R), which provides global variables and functions accessible for both the user interface and server. A more detailed explanation of these three components, a schematic diagram is provided in S1 Fig, and annotated R scripts used for our Shiny app can be found in S1 Data or downloaded from GitHub [34].

To complete a user-defined compositional isotemporal substitution analysis (i.e., to determine the estimated change in health measures associated with user-defined reallocations of time between activities), user input on the initial time allocations and the desired time reallocations is required. Users can specify any time-use reallocation (up to 60 minutes in the current app, but the app developer can set this to any desired duration) provided they do not reallocate more time than what they have available (based on the initial time allocations they have provided). Error messages prompt the user to change their selection if their reallocations would result in a negative value in any of the time-use components. The Shiny app collects the necessary user input and uses the regression coefficients, standard errors and degrees of freedom from the above models to calculate the results for the user-defined reallocation. No unit-level data from the CheckPoint dataset are stored within the app, only the regression coefficients, their standard errors and associated degrees of freedom are stored.

To derive estimates of change in outcome after a user-defined reallocation of time between activities, the Shiny app calculates isometric log-ratio coordinates for the user-specified initial (current) and new (reallocated) composition, and then these coordinates are used as new data inputs for prediction using the stored regression parameters from the models above. To allow the calculation of isometric log-ratio coordinates for compositions where users have allocated zero time to an activity, zero times are replaced with a small duration (3.25 minutes, which is 65% of the 5-minute sampling frame [35]). An alternative solution would be to not allow the user to input any zero durations.

To compare the estimated untransformed health measures (psychosocial health and academic performance) for two activity compositions, the absolute difference between the two estimates is calculated using the corresponding linear combination of the estimated parameters. The 95% confidence interval (CI) for the difference is calculated [36]. Given that the outcome of body fat percentage was log-transformed, the estimated difference in body fat percentage for two activity compositions is calculated using the difference in the exponentiated estimates on the log-scale. For this outcome variable, the 95% confidence interval (CI) for the difference is calculated using the formula of Zou et al [37] to account for lognormal variable transformation and covariance of the two estimates. For ease of interpretation, all estimated differences are also presented as percentage differences.

## Results

The Shiny app web-interface is accessible via a web link [38]. The outputs (estimated differences in health measures for reallocations of time) are based on data from CheckPoint participants with complete health measure and covariate data, and at least one day of time-use recall. Descriptive statistics of the participants can be found in Table 1.

Following adjustment for sex, age, socioeconomic position, and pubertal status the time-use composition was significantly associated with body fat percentage ($F_{27,1637} = 2.66$, $P <$

**Table 1. Sample characteristics (_n_ = 1685).**

| Characteristic | | Participants |
|---|---|---|
| **Sex, n (%)** | | |
| | Male | 869 (51.6) |
| | Female | 816 (48.4) |
| Age (years), mean (SD) | | 12.0 (.4) |
| Socioeconomic z-score, mean (SD) | | .20 (.99) |
| **Pubertal status, n (%)** | | |
| | Pre-pubertal | 127 (9.6) |
| | Early puberty | 330 (25.8) |
| | Mid-pubertal | 665 (51.0) |
| | Late puberty | 167 (13.1) |
| | Post-pubertal | 5 (.5) |
| **24-h activities (min/d), arithmetic mean (SD)** | | |
| | Sleep | 601.7 (59.8) |
| | Screen time | 201.9 (120.8) |
| | Physical activity | 140.1 (83.6) |
| | Quiet time | 82.1 (55.5) |
| | Passive transport | 52.7 (42.6) |
| | School-related | 177.2 (98.0) |
| | Domestic/self-care | 184.4 (71.1) |
| 24-h activities (min/d), compositional means | Sleep; screen time; physical activity; quiet time; passive transport; school-related; domestic/self-care | 709.7; 175.0; 116.9; 70.2; 35.8; 129.1; 203.3 |
| %Body fat, median (IQR) (_n_ = 1672) | | 19.9 (15.5–26.6) |
| Psychosocial health, mean (SD) (_n_ = 1679) | PedsQL Psychosocial Health Summary Score | 77.1 (14.0) |
| Academic performance (writing), mean (SD) (_n_ = 1294) | NAPLAN Writing Component Score | 533.5 (65.7) |

.001), psychosocial health ($F_{27,1644}$ = 4.02, _P_ < .001), and academic performance (writing) ($F_{27,1259}$ = 2.76, _P_ < .001). Please refer to S1 Table for the full set of model parameters.

Reallocating 60 minutes from screen time to physical activity was associated with 4.2% lower body fat (-0.8 [95% CI -1.0 to -0.5] percentage units), 2.5% better psychosocial health (+1.9 [1.4 to 2.5] PedsQL psychosocial health score) and 0.9% higher academic performance (+4.5 [1.8 to 7.2] NAPLAN writing score). Figs 1 and 2 shows how to deduce this information from our app. The first tab (Fig 1, Box 1) on the Shiny app interface (_Initial time use_) allows the user to specify the amount of time they currently spend in each of the seven activities. For the current app, this requires that they vary them from the initial default time-use values which are the sample compositional mean (Fig 1, Box 2). We included an option for the user to provide sex and age (Fig 1, Box 3). Other covariates (e.g., pubertal stage and socioeconomic status z-score) can be included for greater precision when available or if modelling on these parameters is desired. An advanced information tab allows these additional covariates to be provided by the user (Fig 1, Box 4).

In the second tab (Fig 2, Box 1) of the Shiny app (_Specify reallocations_), the user is prompted to define reallocations of their choice. The time-reallocation sliders are shown before any changes have been made (Fig 2, Box 2). The plot on the bottom left (Fig 2, Box 3) shows the starting time-use composition that was specified in the previous tab (_Initial time use)_. On the right (Fig 2, Box 4), the estimated percentage changes in the three health measures are displayed. So far, no time reallocations have been specified; hence the estimated change is zero for all measures.

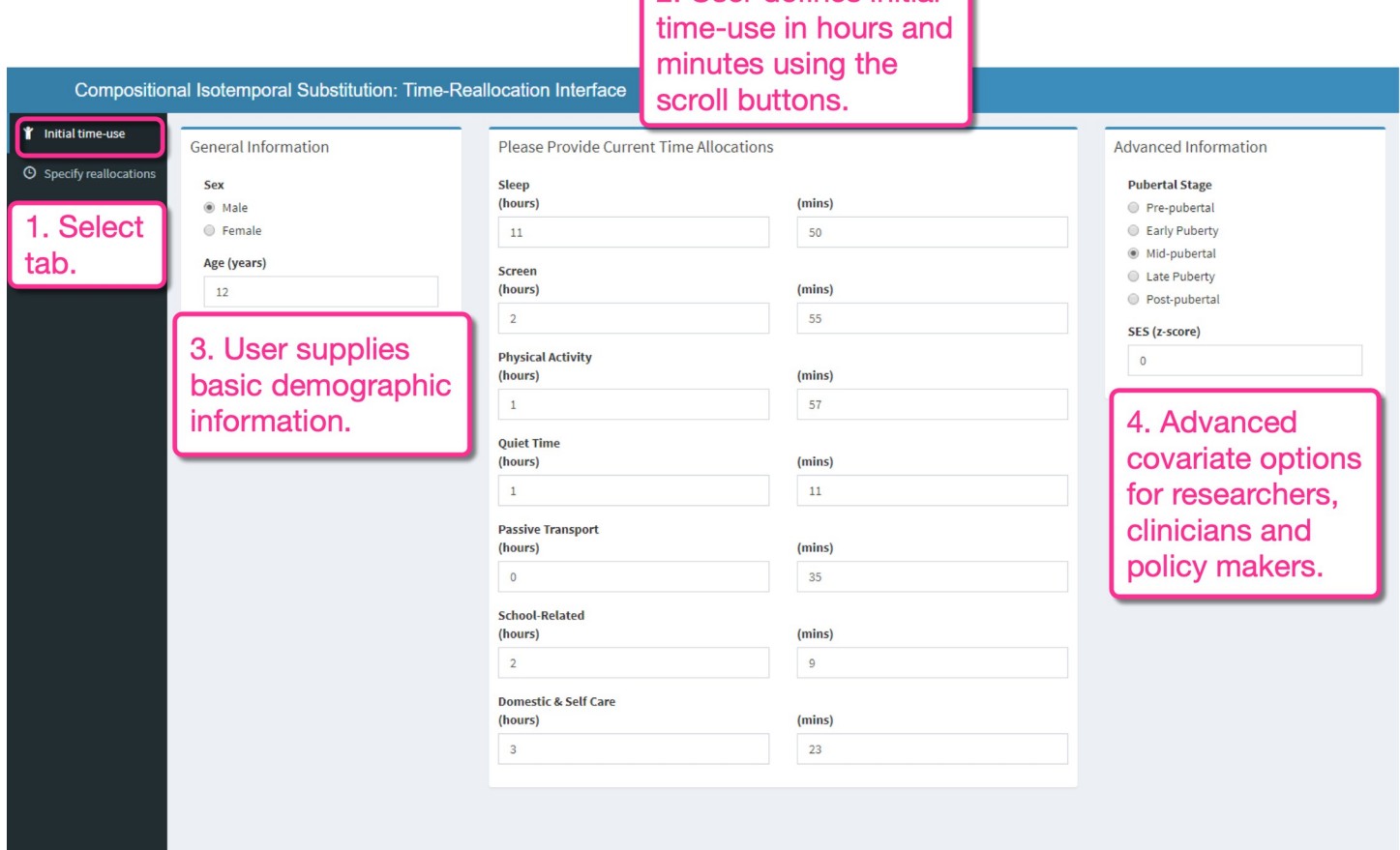

**Fig 1. Shiny app interface.** The initial time-use tab. To use the app to generate estimates for self-selected time reallocations, the user must first specify demographics and initial time-use variables.

Once the "Specify reallocations" tab (Fig 3, Box 1) has been selected, the reallocation of 60 minutes from screen time to physical activity is specified using the sliders (Fig 3, Box 2). The plot on the bottom left shows the new distribution of time across the seven activities (Fig 3, Box 3). On the right, the percent change fields have been updated to show how the specified time reallocation is associated with the difference in health measures (Fig 3, Box 4). The fields are all coloured green to indicate the change is in a favourable direction. Selecting the advanced output reveals two additional plots. The upper plot shows the estimates of the outcomes (with 95% CI) in their raw measurement units, estimated for the initial time-use composition on the left, and for the new time-use composition on the right (Fig 3, Box 5). Below this, there is a plot showing the difference (in raw units) between the two estimates (for the initial and new compositions), with 95% CI (Fig 3, Box 6).

However, when 60 minutes are reallocated to physical activity, the ripple effects across the other activities may be more complex than a one-for-one swap. With a different time-reallocation profile, the associations with health measures may no longer all be beneficial (see Fig 4 for an example). The associations for the complex time reallocation shown in Fig 4 were beneficial for body fat percentage (-4.8%) and psychosocial health (+2.0%), but detrimental for academic performance (-1.3%).

Further, we can see how the estimated differences in health measures may differ depending on the initial time-use composition. The same complex reallocations of time specified in Fig 4 (with

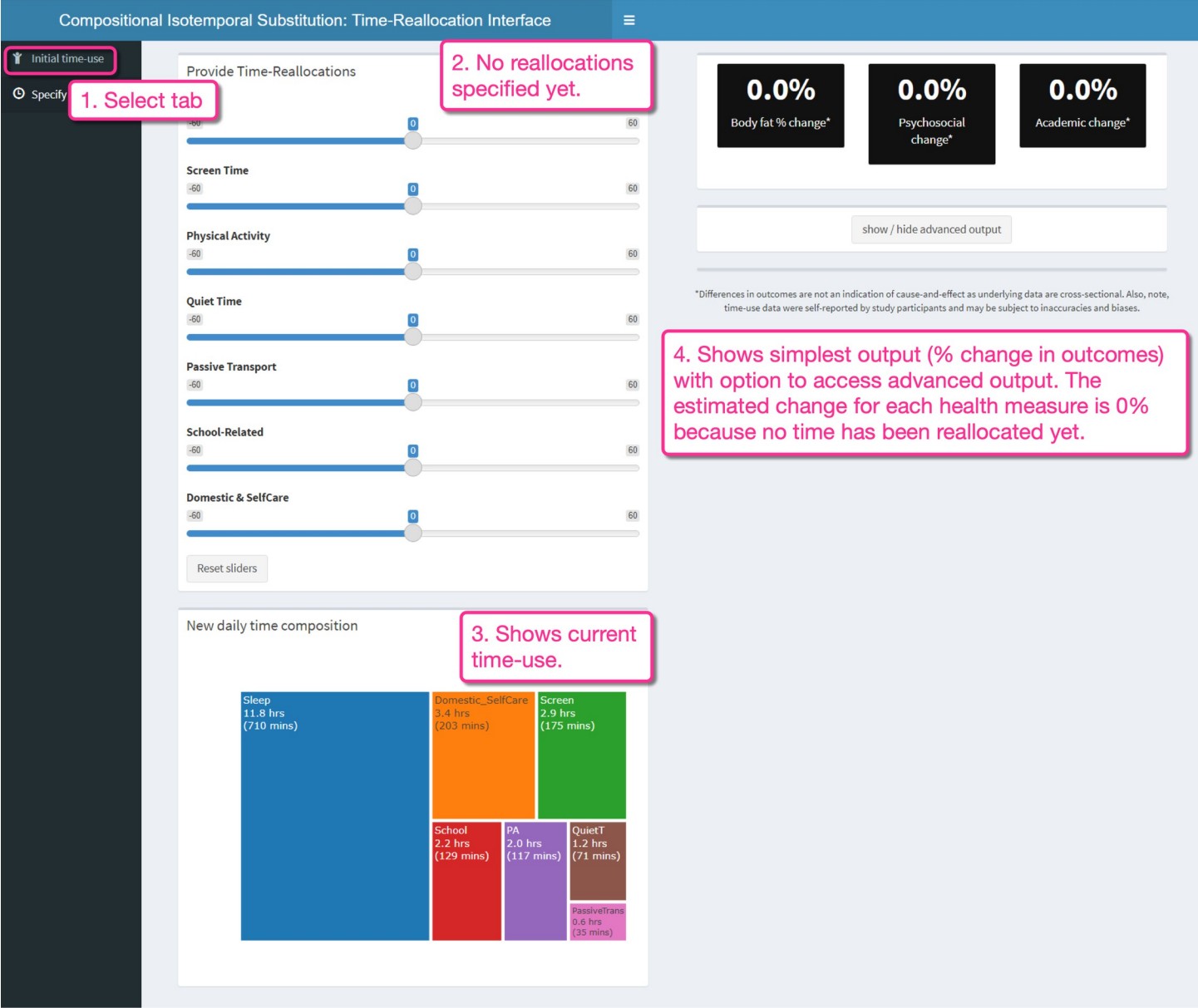

**Fig 2. Shiny app interface.** Time-use reallocation tab.

the compositional mean as the initial time-use composition) were associated with larger differences in body fat (-6.2%) and academic performance (-1.7%) (Fig 5) when a potentially less "healthy" initial composition was specified (1 h less for each of sleep, physical activity and domestic/self-care activities, 1 hour more for each of screen time, quiet time and school related activities).

## Discussion

### Principal results

Our interactive Shiny app provides a simple user interface to explore how health measures are associated with any type of time reallocation from any starting time-use composition

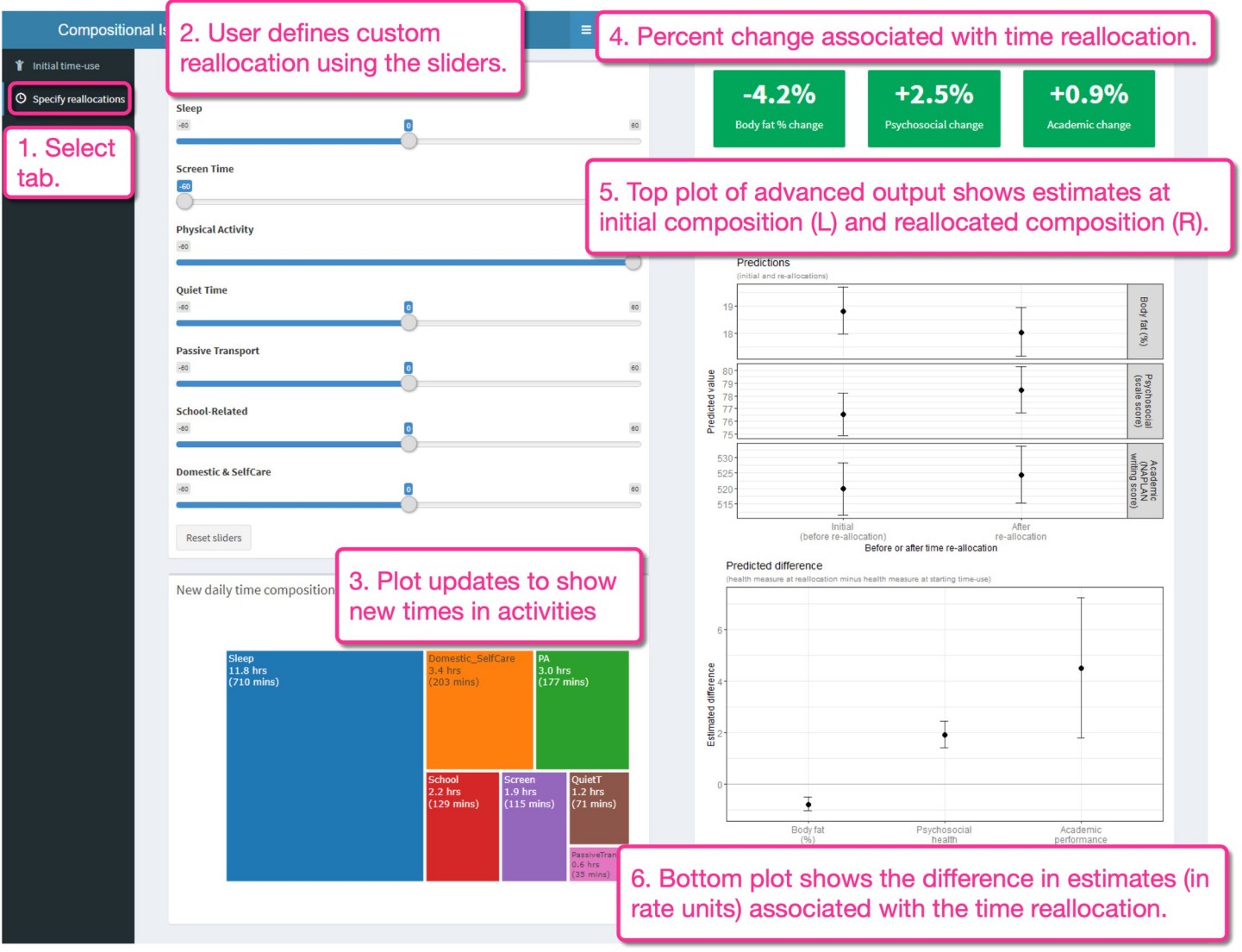

**Fig 3. Specification of custom simple one-for-one time-use reallocation using the app, and visualization of advanced output.** Initial time-use composition was the compositional mean of the sample: Sleep = 11 h 50 min; Screen time = 2 h 55 min; Physical activity = 1 h 57 min; Quiet time = 1 h 11 min; Passive transport = 35 min; School-related activities = 2 h 09 min; Domestic/self-care activities = 3 h 23 min.

commensurate with that of today's 11–12 year olds in population studies. The app enables more accessible and flexible interpretation of the results from compositional regression analyses, which may facilitate translation of research findings to public health promoters, medical practitioners, fitness professionals, policy makers, and the general public. For example, in our illustration of its utility, the app showed that reallocating 60 minutes to physical activity could have diverging associations with health measures depending on which activities the time is drawn from, the initial time-use composition and which health outcome measure is considered.

## Comparison with prior work

Until now, studies have considered only two types of theoretical time reallocations: one-for-one and one-for-remaining. These types of reallocations may not be of interest or

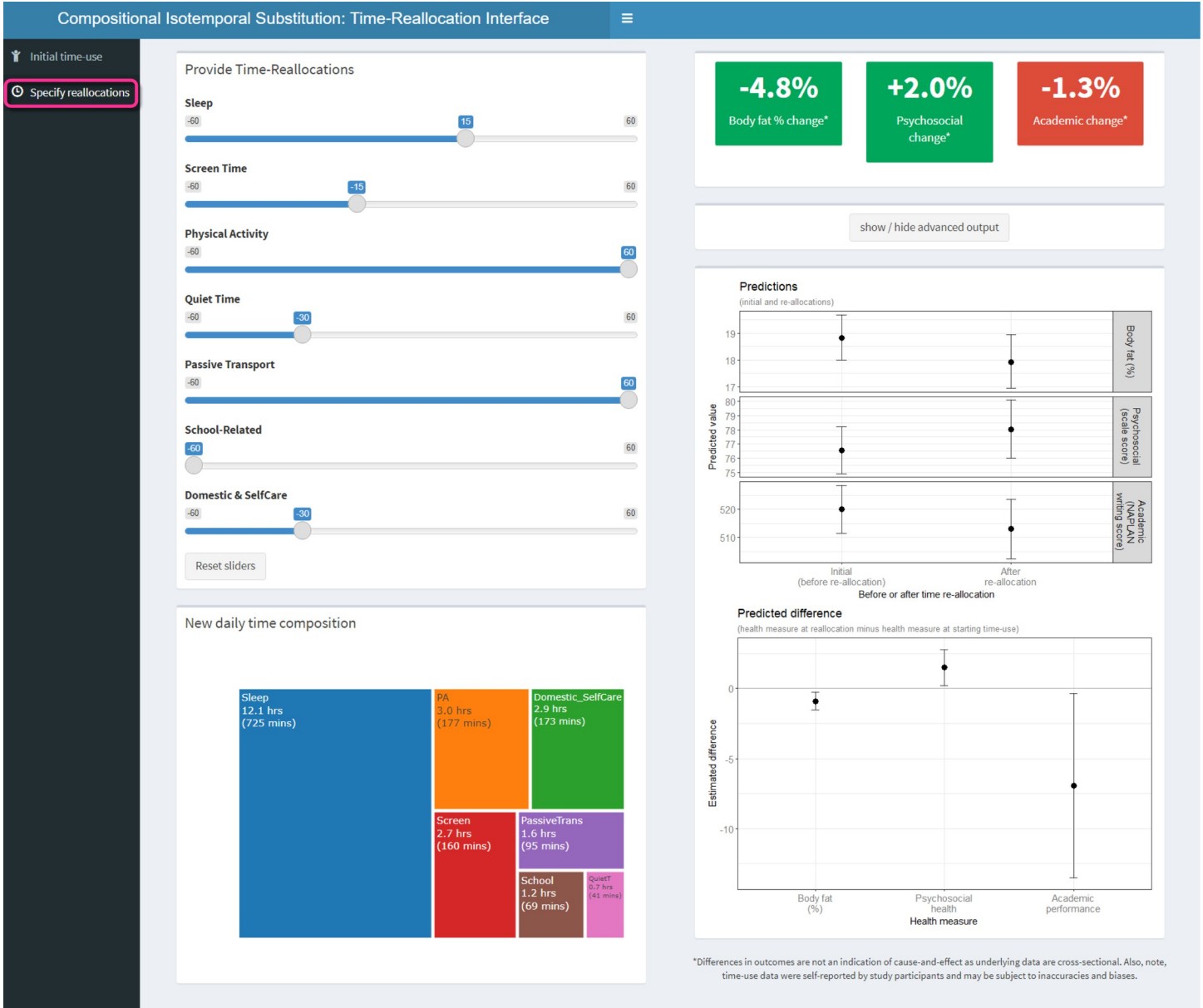

**Fig 4. Specification of complex time-use reallocations using the app, and visualization of advanced output.** Initial time-use composition was the compositional mean of the sample: Sleep = 11 h 50 min; Screen time = 2 h 55 min; Physical activity = 1 h 57 min; Quiet time = 1 h 11 min; Passive transport = 35 min; School-related activities = 2 h 09 min; Domestic/self-care activities = 3 h 23 min.

representative of how people actually reallocate their time, that is, *empirical* reallocations. To our knowledge, our Shiny app is the first tool to provide the user with the opportunity to explore the health associations of their own or their clients' selected time reallocations. In our example, manipulation of the sliders confirmed previous research findings of better body composition and psychosocial health being associated with higher reallocations of time to physical activity, particularly by reducing the time spent in sedentary activities (screen time, quiet time) [39–41]. Interestingly, reallocations to physical activity did not seem to be as beneficial for academic performance. As found in this study, previous research also suggests that the benefit of

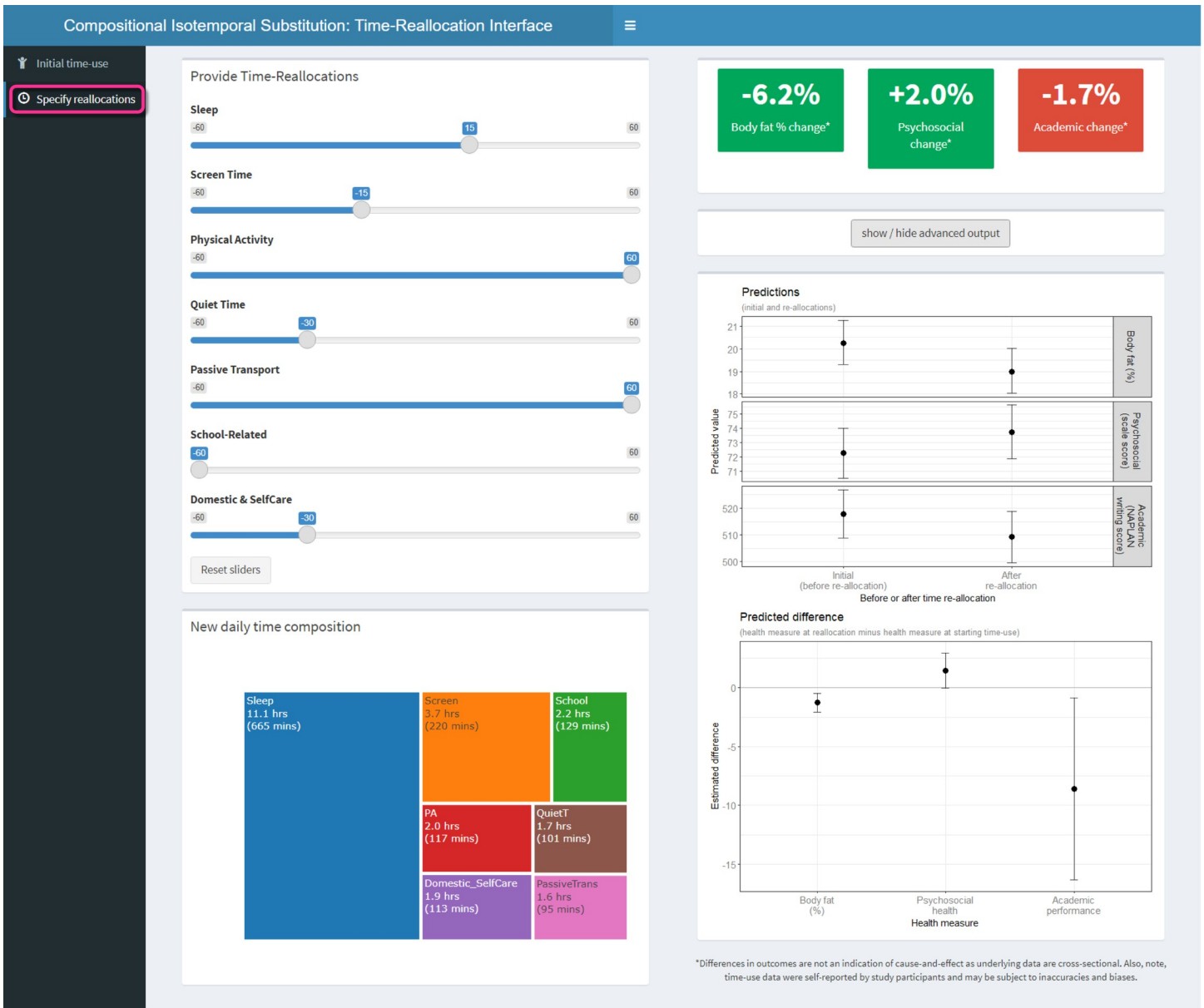

**Fig 5. Specification of the same complex reallocation as in Fig 4, but relative to a different initial composition where Sleep = 10 h 50 min; Screen time = 3 h 55 min; Physical activity = 57 min; Quiet time = 2 h 11 min; Passive transport = 35 min; School-related activities = 3 h 09 min; Domestic/self-care activities = 2 h 23 min.**

reallocations to physical activity differs depending on which activities the time is drawn from, and on the starting durations of the activities [17].

## Strengths and limitations

Strengths of this study include a newly developed Shiny app allowing users to interact with the research findings themselves, exploring time-reallocation scenarios that are of interest to them or their clients. The user-friendly interface of the Shiny app enables usage not only by researchers but also by public health promoters, medical practitioners, fitness professionals, policy makers, and members of general public. The simple customization or personalization

enabled by our Shiny app may facilitate their engagement with, and understanding of, the research findings. Another strength of our Shiny app is that all underlying computations are derived from summary statistics without requiring the user to have direct access to individual-level data, overcoming data confidentiality issues. Iterative improvements can be made to the app's interface and underlying modelling/data, by updating the associated open source R scripts.

We used cross-sectional data, meaning that causation cannot be inferred, that is, we cannot be certain that the time-use reallocation is causing the difference in health, and therefore that an individual will experience what is predicted by the app. Future iterations of the app should provide the user with the opportunity to click on links that provide background information regarding the data source (e.g., study design and location, number and demographics of participants) and important considerations for interpreting the results (e.g., that causation cannot be inferred). Our current app includes a brief disclaimer regarding the cross-sectional nature of the data, and that the activity data were derived from self-reports. Although the MARCA instrument has good-to-excellent measurement properties for children of this age [21], there remains the potential of reporter bias or recall error. However, device-based options for large scale time-use measurement (e.g. accelerometry) cannot yet differentiate all categories meaningful to participants within intensity groupings (e.g. different types of sedentary behavior). Without completing the MARCA or similar assessment, the user is unlikely to accurately recall their own time use when completing the "initial time use" information, but this could be improved in future versions of the app by providing the option to complete a time-use recall tool which feeds directly into the "initial time use" tab. Care must be taken when generalising findings. The underlying data are from sample who have a slightly higher socioeconomic advantage and homogeneity on a census-derived composite index at postcode level (SEIFA) [42] (mean 1028, SD 60 compared to national mean 1000, SD 100) as the general Australian population. This Shiny app prototype is restricted to a narrow age band (11–12 years), and it does not separately consider intersex children and children who identify their gender as non-binary. To be able to overcome the latter limitation, the modelling may require a very large underlying sample that collects this potentially sensitive information.

## Implications for practice

Our Shiny app interface is the first to allow customized estimation of the impact of time reallocations. Future iterations of the app could provide a way of storing the reallocations a user has tried, and returning a printable report at the end of the session. The app may empower individuals and their supporting professionals/policy makers to make informed lifestyle choices according to what they value. It is designed to promote flexibility, autonomy and self-determination, which behaviour-change theory and empirical research show to be crucial to sustained lifestyle improvement [43, 44]. By guiding the user to plan and be deliberate about their behaviour-change goals, the app may foster a sense of commitment and engagement. Working with relevant end users through feasibility studies exploring the utility and accessibility of the app, future iterations can be tailored to better suit a variety of clinical and population users. We have illustrated the utility of the app using cross-sectional data; however, the same type of interface could be used for longitudinal or intervention data, and for a wider range of health and wellbeing measures such as cardiovascular and metabolic outcomes or depressive symptoms. Future longitudinal studies may explore how people actually reallocate time when their everyday activity balance is disrupted–empirical reallocations—and the socio-demographic factors and starting activity compositions associated with different reallocation "choices". The app could be used to generate research hypotheses about the effect of time reallocations which

can be tested in randomised controlled trials. Large trials of digitally enabled cohorts (such as the GenV initiative now recruiting in Victoria, Australia [45]) could provide opportunities for randomised trials to test efficacy of a further-refined app in achieving behaviour change and health gains. Interactive interfaces for models based on big data may allow personalized precision-type estimation, providing choice and ownership to an individual seeking to set lifestyle goals to improve their health. Apps based on such data will enable customization of future interventions, and contribute to health-economic modelling of these interventions.

## Conclusions

The Shiny app developed in this study allows the health associations of various time reallocations to be easily explored and compared against each other. This provides future users with autonomy in investigating lifestyle changes of their interest, which are relevant and meaningful to them or their clients. As a translation tool, the interactive interface showcased in this study has the potential to transform the way research findings are disseminated to public health promoters, medical practitioners, fitness professionals, policy makers, and the general public. Ultimately, making research findings accessible will enable people to make choices about their own lives, and enable professionals to make evidence-based recommendations for others' time use.

## Supporting information

**S1 Fig. Shiny app detail and schematic diagram.**
(PDF)

**S1 Table. Compositional model summaries.**
(PDF)

**S1 Data. Shiny app code.**
(PDF)

**S1 File.**
(PDF)

## Acknowledgments

This paper uses unit record data from Growing Up in Australia, the Longitudinal Study of Australian Children. The study is conducted in partnership between DSS, the Australian Institute of Family Studies (AIFS) and the Australian Bureau of Statistics (ABS). We thank the LSAC and CheckPoint study participants, staff and students for their contributions. The findings and views reported in this paper are solely those of the authors and should not be attributed to DSS, AIFS or the ABS. Dorothea Dumuid and Ty Stanford (University of South Australia) had full access to all the data in the study and take responsibility for the integrity of the data and the accuracy of the data analysis. REDCap (Research Electronic Data Capture) electronic data capture tools were used in this study. More information about this software can be found at: www.project-redcap.org.

## Author Contributions

**Conceptualization:** Dorothea Dumuid, Timothy Olds, Željko Pedišić, David JR. Foster, Andrew J. Atkin, Leon Straker, Francois Fraysse, Ross T. Smith, Frank Neumann, Ron S. Kenett, Paul Jarle Mork.

**Data curation:** Timothy Olds, Melissa Wake.

**Formal analysis:** Dorothea Dumuid, Timothy Olds, Ty Stanford.

**Funding acquisition:** Dorothea Dumuid, Timothy Olds, Melissa Wake, Željko Pedišić, David JR. Foster, Andrew J. Atkin, Leon Straker, Francois Fraysse, Ross T. Smith, Frank Neumann, Ron S. Kenett, Paul Jarle Mork.

**Investigation:** Dorothea Dumuid, Timothy Olds, Melissa Wake, Charlotte Lund Rasmussen, Željko Pedišić, Jim H. Hughes, David JR. Foster, Rosemary Walmsley, Andrew J. Atkin, Leon Straker, Francois Fraysse, Ross T. Smith, Frank Neumann, Ron S. Kenett, Paul Jarle Mork, Derrick Bennett, Aiden Doherty, Ty Stanford.

**Methodology:** Dorothea Dumuid, Melissa Wake, Charlotte Lund Rasmussen, Željko Pedišić, Jim H. Hughes, David JR. Foster, Rosemary Walmsley, Andrew J. Atkin, Leon Straker, Francois Fraysse, Ross T. Smith, Frank Neumann, Ron S. Kenett, Paul Jarle Mork, Derrick Bennett, Aiden Doherty, Ty Stanford.

**Project administration:** Melissa Wake, Francois Fraysse, Ross T. Smith, Frank Neumann.

**Resources:** Melissa Wake.

**Software:** Dorothea Dumuid, Jim H. Hughes, Ty Stanford.

**Visualization:** Dorothea Dumuid, Charlotte Lund Rasmussen, Rosemary Walmsley, Derrick Bennett, Aiden Doherty, Ty Stanford.

**Writing – original draft:** Dorothea Dumuid, Timothy Olds, Melissa Wake, Charlotte Lund Rasmussen, Željko Pedišić, Jim H. Hughes, David JR. Foster, Rosemary Walmsley, Andrew J. Atkin, Leon Straker, Francois Fraysse, Ross T. Smith, Frank Neumann, Ron S. Kenett, Paul Jarle Mork, Derrick Bennett, Aiden Doherty, Ty Stanford.

**Writing – review & editing:** Dorothea Dumuid, Timothy Olds, Melissa Wake, Charlotte Lund Rasmussen, Željko Pedišić, Jim H. Hughes, David JR. Foster, Rosemary Walmsley, Andrew J. Atkin, Leon Straker, Francois Fraysse, Ross T. Smith, Frank Neumann, Ron S. Kenett, Paul Jarle Mork, Derrick Bennett, Aiden Doherty, Ty Stanford.

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
