## [Decision Letter · Decision Letter 0]

21 Mar 2022

PONE-D-21-36238Your Best Day: An interactive app to translate how time reallocations within a 24-hour day are associated with health measuresPLOS ONE

Dear Dr. Dumuid,

Thank you for submitting your manuscript to PLOS ONE. After careful consideration, we feel that it has merit but does not fully meet PLOS ONE’s publication criteria as it currently stands. Therefore, we invite you to submit a revised version of the manuscript that addresses the points raised during the review process.

Please address the reviewer's comments with special emphasis placed on:1. Discussion of the health improvement mechanism drivers2. Selection of the covariates used in the models3. Differentiation between the association and  the causation

We look forward to receiving your revised manuscript.

Kind regards,

Jaroslaw Harezlak, PhD

Academic Editor

PLOS ONE

Reviewers' comments:

Reviewer's Responses to Questions

**Comments to the Author**

1. Is the manuscript technically sound, and do the data support the conclusions?

Reviewer #1: Yes

Reviewer #2: Yes

2. Has the statistical analysis been performed appropriately and rigorously? 

Reviewer #1: Yes

Reviewer #2: I Don't Know

3. Have the authors made all data underlying the findings in their manuscript fully available?

Reviewer #1: Yes

Reviewer #2: Yes

4. Is the manuscript presented in an intelligible fashion and written in standard English?

Reviewer #1: Yes

Reviewer #2: Yes

5. Review Comments to the Author

Reviewer #1: This paper has important findings for user-defined time-use reallocations upon health outcomes via an app intervention. The paper finds that such reallocations lead to better objective and subjective health outcomes. In light of the fact that reallocations are user driven, app driven and continuous in nature, the results are novel and promising for users.

On pages 5-6 of the manuscript, the point (empirical strategy) is made that log ratio form explanatory variables help to overcome perfect multicollinearity in the model (i.e., activity_1_time +…+ activity_t_time = 24 (hours)). It is then stated that log ratio form coefficients are more difficult to interpret within the model. Given this drawback, I suggest that the authors run an alternative model that uses the empirical strategy of leaving one category dummy variable out of the specification (i.e., create a reference group). In this manner, all coefficients are interpreted relative to the reference group. This would allow an alternative specification that maintains coefficient interpretability while also avoiding the dummy variable trap.

I would like more discussion of what mechanism is driving the health improvements (e.g., based on past results from the literature). Is it the salience of seeing scheduled time use reallocations within an app that is driving behavioral change? Does the schedule create a commitment device that causes the behavioral change? Is it the act of planning and being deliberate in your time use decisions?

Reviewer #2: Dumuid et al. developed a web application so that users can easily estimate the association between time reallocations among various activities on body fat percentage, psychosocial health, and academic performance on a writing test in children, estimated using the compositional isotemporal substitution model. The manuscript is very well written and easy to follow the project goals and logic, and limitations are adequately discussed. The authors are careful in their use of language to not imply causal relationships from this method. The user-friendly open web application that produces estimates in interpretable outputs appears novel, and will make it easier to translate the compositional isotemporal substitution method to other outcomes and datasets.

In the dataset used to produce results in the application, a strength is that body fat percentage was measured (not self-report), and academic performance and psychosocial health assessed from standardized test/survey. 24h recalls were used to obtain self-report activity estimates from the children. The authors note this limitation in the discussion, which can be overcome with technological iterations and incorporated into the app in the future. I have downloaded the RShiny app and ran it locally to confirm that it works, and tested the web version as well. The directions were clear and it is easy to operate.

Minor comments:

Covariates were self-reported (from child or parent). How did the authors decide which covariates to include in models?

The authors note in their discussion that the application can be helpful to public health promoters, medical practitioners, policy makers, etc. To push toward establishing causal relationships, this also seems useful to easily generate research hypotheses to test in interventions (at least the less complex allocations may be feasible to test). Given the limitations of traditional RCTs which would be impossible to assess all combinations of allocations, perhaps alternative designs could be used. There would be many challenges, but given a hypothesis generated by the application, perhaps in certain contexts, one may be able to use designs such as a micro-randomized trial to tease out the relative effectiveness of time reallocation scenarios in context of how time is already allocated. Reallocations could be randomized based on self-reported activities from mobile phones. Do the authors have thoughts on this use of their application?

The authors appropriately note the issue of causation in their limitations and that future iterations of the app should explicitly warn users to take caution in how they interpret results. I suggest that the authors add this now to the app, because assuming causation from assocational data is very common in the context of public health and policy, and given that they are among the targets of the application, this seems like an easy addition to help prevent misinterpretations going forward. Some notes on the self-reported nature of the activities seems warranted as well, or at least a prominent message and link to the paper once published to refer them to the methods.

The figure quality makes them difficult to read even after downloading the files in their original .tiff format.

6. PLOS authors have the option to publish the peer review history of their article (what does this mean?). If published, this will include your full peer review and any attached files.

Reviewer #1: **Yes: **Bhavneet Walia

Reviewer #2: No

---

## [Author Response · Author response to Decision Letter 0]

18 May 2022

Dear reviewers, 

thank you for your positive feedback and constructive comments. We have provided a point-by-point response to these comments in the "Response to Reviewers" document.

Kind regards,

Dot Dumuid, on behalf of all co-authors.

---

## [Decision Letter · Decision Letter 1]

19 Jul 2022

Your Best Day: An interactive app to translate how time reallocations within a 24-hour day are associated with health measures

PONE-D-21-36238R1

Dear Dr. Dumuid,

We’re pleased to inform you that your manuscript has been judged scientifically suitable for publication and will be formally accepted for publication once it meets all outstanding technical requirements.

Kind regards,

Jaroslaw Harezlak, PhD

Academic Editor

PLOS ONE

Additional Editor Comments (optional):

Reviewers' comments:

Reviewer's Responses to Questions

**Comments to the Author**

1. If the authors have adequately addressed your comments raised in a previous round of review and you feel that this manuscript is now acceptable for publication, you may indicate that here to bypass the “Comments to the Author” section, enter your conflict of interest statement in the “Confidential to Editor” section, and submit your "Accept" recommendation.

Reviewer #1: All comments have been addressed

Reviewer #2: All comments have been addressed

2. Is the manuscript technically sound, and do the data support the conclusions?

Reviewer #1: Yes

Reviewer #2: Yes

3. Has the statistical analysis been performed appropriately and rigorously? 

Reviewer #1: Yes

Reviewer #2: I Don't Know

4. Have the authors made all data underlying the findings in their manuscript fully available?

Reviewer #1: Yes

Reviewer #2: No

5. Is the manuscript presented in an intelligible fashion and written in standard English?

Reviewer #1: Yes

Reviewer #2: Yes

6. Review Comments to the Author

Reviewer #1: The authors have addressed all comments and suggestions satisfactorily. The paper now represents a strong addition to the literature.

Reviewer #2: (No Response)

7. PLOS authors have the option to publish the peer review history of their article (what does this mean?). If published, this will include your full peer review and any attached files.

Reviewer #1: No

Reviewer #2: **Yes: **Colby Vorland

---

## [Editor Report · Acceptance letter]

11 Aug 2022

PONE-D-21-36238R1 

Your Best Day: An interactive app to translate how time reallocations within a 24-hour day are associated with health measures 

Dear Dr. Dumuid:

I'm pleased to inform you that your manuscript has been deemed suitable for publication in PLOS ONE. Congratulations! Your manuscript is now with our production department. 

Kind regards, 

on behalf of

Dr. Jaroslaw Harezlak 

Academic Editor

PLOS ONE